# Routine Mitogenome MPS Analysis from 1 and 5 mm of Rootless Human Hair

**DOI:** 10.3390/genes13112144

**Published:** 2022-11-18

**Authors:** Lauren C. Canale, Jennifer A. McElhoe, Gloria Dimick, Katherine M. DeHeer, Jason Beckert, Mitchell M. Holland

**Affiliations:** 1Forensic Science Program, Department of Biochemistry & Molecular Biology, Eberly College of Science, Pennsylvania State University, University Park, PA 16802, USA; 2Mitotyping Technologies, 2565 Park Center Blvd., Suite 200, State College, PA 16801, USA; 3Indiana State Police, 100 North Senate Avenue, Indianapolis, IN 46204, USA; 4Microtrace 790 Fletcher Drive, Suite 106, Elgin, IL 60123, USA

**Keywords:** next generation sequencing, forensic science, hair

## Abstract

While hair shafts are a common evidence type in forensic cases, they are often excluded from DNA analysis due to their limited DNA quantity and quality. Mitochondrial (mt) DNA sequencing is the method of choice when working with rootless hair shaft fragments due to the elevated copy number of mtDNA and the highly degraded nature of nuclear (n) DNA. Using massively parallel sequencing (MPS) of the mitochondrial (mito) genome, we studied the impact of hair age (time since collection) and physical characteristics (hair diameter, medullary structure, and length of hair tested) on mtDNA recovery and MPS data quality. Hair shaft cuttings of 1 and 5 mm from hairs less than five years to 46 years of age from 60 donors were characterized microscopically. Mitogenome sequences were generated using the Promega PowerSeq^TM^ Whole Mito System prototype kit and the Illumina MiSeq instrument. Reportable mitogenome sequences were obtained from all hairs up to 27 years of age (37 donors), with at least 98% of the mitogenome reported for more than 94% of the 74 hair samples analyzed; the minimum reported sequence was 88%. Furthermore, data from the 1 and 5 mm replicates gave concordant haplotypes. As expected, mtDNA yield decreased, mtDNA degradation increased, and mitogenome MPS data quality declined as the age of the hair increased. Hair diameter and medullary structure had minimal impact on yield and data quality. Our findings support that MPS is a robust and reliable method for routinely generating mitogenome sequences from 1 and 5 mm hair shaft samples up to 27 years of age, which is of interest to the forensic community, biological anthropologists, and medical geneticists.

## 1. Introduction

Hairs are one of the most common evidence types found at crime scenes [1,2,3] and are a useful sample type for medical research [4,5,6] and archaeological investigations [7,8]. Nuclear (n) and mitochondrial (mt) DNA can both be recovered from hairs. For example, hairs in the anagen and catagen growth stages present follicular tissue, especially when forcibly removed, allowing for nDNA analysis when roots are present. Telogen hairs are easily shed, but due to the keratinization process leading to the degradation of cellular organelles and chromosomal nucleic acids, nDNA is generally too degraded for conventional analysis [9,10,11]. However, under certain circumstances, nDNA can be recovered and analyzed from hair shafts [10,12,13,14]. Along the shafts of hairs in all phases, nDNA is present in higher quantities than mtDNA, but due to its extensive fragmentation is typically not of sufficient quality for amplicon-based nDNA analysis; for example, short tandem repeat (STR) profiling [10]. Therefore, mtDNA sequencing remains the method of choice when analyzing shed hairs or small fragments of hair shafts.

Historically, mtDNA testing has focused on recovering the control region (CR) of the mitogenome using electrophoretic techniques such as Sanger type sequencing (STS) [15]. Haplotypes generated from the CR are highly informative, however, expanding to the entire mitogenome significantly increases the discriminatory power [16]. Unfortunately, sequencing the mitogenome using STS is labor-intensive, requires large quantities of DNA, and is a low-throughput method. Contrary to this, with advancements in massively parallel sequencing (MPS), analysis of the mitogenome can be performed relatively quickly with higher throughput and on samples that are routinely encountered in forensic cases, including hair shafts [17].

Microscopical analysis of hair has historically been an integral part of forensic investigations and remains an important component of hair analysis [18,19]. Hair characterization typically involves an assessment of color, size, distribution and pattern of pigment granules, the appearance and structure of the cuticle, cortex, and medulla, indications of chemical treatments, the physical condition of the shaft including diameter, and other microscopical characteristics [20,21]. Comparison to reference hairs and assessment of inclusion or exclusion remains a relatively subjective process [18]. Currently, hairs are assessed microscopically prior to DNA analysis, noting if a root is present and the growth phase of the root to determine whether nDNA profiling or mtDNA sequencing should be performed. Crime laboratories that do not perform mtDNA sequencing will either not move forward with these hairs or send them to a private laboratory that does sequence analysis.

Previous research has focused on either the mtDNA CR or on small sample sizes when performing mitogenome analysis [11,22]. In addition, the majority of the tested hairs have been longer and/or recently collected with a little background on microscopic characteristics or hair treatments. Gallimore et al. [11] observed heteroplasmy in the CR using an MPS approach in 75 hair shafts from five donors that were all tested shortly after collection and were 2 cm cuttings. Parson et al. [22] successfully recovered complete mitogenomes from hair shafts and shed hairs but tested a small sample size of ten hairs. A survey study assessing 691 casework hairs analyzed using an STS approach over many years drew comparisons between any noted physical characteristics with the resulting CR (HV1/HV2) sequence profiles [1]. They concluded that with increasing age, the likelihood of generating a full sequence profile decreased, while with increasing natural hair color and diameter, the likelihood increased. To our knowledge, a similar study has not been done using an MPS approach on the entire mitogenome.

Microscopical analysis of hairs prior to sequencing is typically not performed to assess whether morphological features contribute to the outcome. In addition, forensic laboratories often receive small hair fragments for analysis (0.1–2+ cm), with 2 cm being the typical sample size tested. Hair evidence can also be stored for years if part of a cold or historical case. The goal of the current study was to assess the quantity and quality of mitogenome MPS data from hair samples representative of those encountered in crime laboratories when considering age (since time of collection) and physical characteristics of the tested hairs (for example, hair diameter, medullary structure, and length of hair tested).

## 2. Materials and Methods

### 2.1. Sample Collection and Preparation

Head hair samples from 60 donors were collected from two operating forensic laboratories (Mitotyping Technologies, LLC, a Division of SoftGenetics, State College, PA, USA and Microtrace, LLC, Elgin, IL, USA) and an academic research laboratory (Pennsylvania State University, University Park, PA, USA). The donated hairs fell into three age categories for time since removal from head: 13 recent (R) hairs (<5 years), 24 old (O) hairs (5–27 years), and 23 older (VO) hairs (41–46 years). Twelve of the R hair samples were closer to five years from collection. The 24 O hairs had an average age of almost 14 years. Appendix A, provides the age for each hair sample.

Twelve of the R hair samples were donated for a previous hair study [11], under the Penn State University internal review board (IRB) approved protocol STUDY00000970, and one was collected under STUDY00014305. These hairs were stored in paper envelopes at room temperature. Twelve of the O hair samples were provided by Mitotyping and were stored at −20 °C as part of proficiency testing kits. Once received from Mitotyping, the hairs were stored in paper envelopes at room temperature for approximately one month before MPS analysis. The remaining hair samples (12 O hairs and 23 VO hairs) were provided by Microtrace and were stored in plastic bags at room temperature. Once received from Microtrace, the hairs were stored in plastic bags at room temperature for approximately five days before MPS analysis.

Two cuttings were sampled from each hair, 1 mm and 5 mm in length. Cuttings were taken 1 cm from the root end when a root was present (45 of the 60 hairs) and from the proximal end at an unknown distance from the root when it was not present. The 120 cuttings were then subjected to microscopical analyses.

### 2.2. Microscopy

The 1 and 5 mm cuttings were analyzed using a Leica FS4000 microscope at 200× magnification. The hairs were mounted on glass microscope slides using deionized water, and photographs were taken using an MU 1000 camera and AmScope software version 3.7. Figure 1 provides representative pictures of hairs from the study. Microscopic characteristics noted were the width, the presence or absence of a medulla, and the pigmentation. The width of the hairs was measured by calibrating the AmScope software with a micrometer. If the width of the hair changed, two measurements were taken, one at the largest diameter and one at the smallest diameter, and the average was taken and used for data analysis. If a medulla was present, it was categorized as either trace, discontinuous, or continuous. Hair treatment information was mostly unknown for the hair samples, except for the hairs received from Microtrace. All microscopy data can be found in Appendix A.

### 2.3. DNA Extraction

Extraction of DNA from each hair cutting was conducted using the method reported by Gallimore et al. [11]. Cuttings were first cleaned in 1 mL of 5% Terg-a-zyme for 20 min with sonication. Duplicate washes of 1 mL 100% ethanol and 1 mL MGW water were performed to remove the Terg-a-zyme. A 300 μL aliquot of ATL buffer (Qiagen; Hilden, Germany), 20 μL of 1 M DTT, and 20 μg/mL Pro K were added to the clean cuttings and then incubated in a thermomixer (Eppendorf; Hamburg, Germany) at 56 °C and 900 RPM for one hour, followed by the addition of 300 μL of AL buffer (Qiagen; Hilden, Germany) and a second incubation at 70 °C and 900 RPM for 10 min. Lysates were purified using the PrepFiler^®^ Forensic DNA Extraction Kit (Applied Biosystems; Foster City, CA, USA) according to the manufacturers protocol with the following exceptions: 300 μL of isopropanol and 40 μL of elution buffer according to Gallimore et al. [11]. Modifications to the Gallimore et al. protocol included a second water wash during cleaning and the addition of a second thermomixer for room temperature incubation. Twenty hair cuttings were extracted at a time with one reagent blank.

Twenty-one (21) of the 120 hair cuttings (two R, four O, and 15 VO) were re-extracted and the testing was repeated due to suspected contamination based on mixed sites at major SNP (single nucleotide polymorphism) sites. Eleven (11) of the hairs had presumed contamination from laboratory personnel and the remaining 10 hairs had possible contamination from an unknown source. The additional testing addressed any suspected contamination concerns, and as a result, there was no effect on the findings and conclusions presented in this study.

### 2.4. mtqPCR Assay

A custom mtDNA quantitative PCR assay (mtqPCR) described in Gallimore et al. [11] was used to quantify copies of mtDNA recovered per μL of DNA extract. The eight concentrations for the standard curve were 200,000, 20,000, 2000, 200, 150, 100, 50, and 10 mtDNA copies/μL. Copies of mtDNA per unit volume in each hair extract were quantified immediately following extraction. All quantifications were performed on an Applied Biosystems^®^ 7500 instrument (Foster City, CA, USA) and performed in duplicate.

### 2.5. Amplification of the Mitogenome, Library Preparation, and MiSeq Conditions

Each hair extract was amplified using the PowerSeq^TM^ Whole Mito System prototype kit from Promega (Madison, WI, USA), targeting an input of 40,000 copies of mtDNA or the maximum extract volume of 15 μL for low-yield samples, with partial MPS results obtained from as few as 110 copies of mtDNA template (Appendix A). The kit provides for a single or dual multiplexed PCR reaction with 161 amplicons covering the mitogenome. The single multiplexed approach was used for the samples in this study. A maximum of 23 samples were included per amplification and library preparation run (20 hair extracts, one reagent blank, one negative amplification control, and one positive amplification control) except for the final run, which included 21 hair extracts for a total of 24 samples. Amplification, purification, quantification, and normalization were according to the manufacturer’s protocol. Library preparation was performed with the Illumina^®^ TruSeq DNA PCR-Free HT kit and IDT for Illumina^®^ TruSeq DNA UD Indexes (Illumina; San Diego, CA, USA). If samples did not meet the recommended 4 nM library concentration, a concentration of 2 nM was used for pooling. Samples that did not meet the 2 nM were not diluted prior to pooling. Two runs were pooled at 1.5 nM due to most samples being below 2 nM. The pooled libraries were spiked with 10% PhiX with a final loading concentration of approximately 7.0 pM. The prepared libraries were run on the Illumina^®^ MiSeq with a 600-cycle v3 kit (Illumina; San Diego, CA, USA), 276 × 276 paired read ends.

### 2.6. MPS Data Analysis

The fastq files generated from the MiSeq Reporter software were analyzed using GeneMarker^®^ HTS (SoftGenetics; State College, PA, USA), version 2.2.5.0 [23]; referred to as GM HTS. The sequences were aligned to the revised Cambridge Reference Sequence (rCRS) [24] with a custom motif file to ensure phylogenetically correct calls. SNPs present in more than 50% of the reads were considered major variants and referred to as the haplotype. Minor variants are SNPs that are in less than 50% of the reads. When assessing the minor variants, all indels (insertion/deletion) were excluded along with the following variants commonly seen when using the PowerSeq Whole Mito System: 523A, 524C, 663G, 709G, 2706A, 3010G, 7028C, 8019T, 8860A, 13760T, and 15326A. The filter settings included a minimum read coverage of 200 for the haplotype, a minimum of 40 reads for reporting minor variants, a minimum threshold of 2% for reporting minor variants, ≤10 for the allele score difference, ≤2.5 for the SNP balance ratio, and ≤5.0 for the indel balance ratio. Haplogroups were generated for each sample using HaploGrep2 (version 2.2). Sequences generated from the VO hairs contained significantly more noise in the MPS data (see Section 3.3); measured as the number of minor sites reported. As a result, a detailed analysis of the VO hairs will be the subject of a future study.

### 2.7. Statistical Analysis

Normality was tested using the Shapiro–Wilk test [25], which showed the dataset was non-normal. Therefore, the non-parametric Kruskal–Wallis test [26] with a Dunn’s post hoc test [27] utilizing the Holm correction for multiple comparison testing [28] was used to evaluate the statistical significance of differences in the analyses of mtDNA yield, the MPS data quality, and the microscopical correlations (critical *p*-value < 0.05). The analysis of the total error rate per sample was performed using a custom R script generated for a previous study [29] and significance was also calculated using the Kruskal–Wallis test with a Dunn’s post hoc test adjusted with the Holm correction (critical *p*-value < 0.05). A simple linear regression model was run to determine if the mtDNA yield could be predicted by the width of the hairs and if the MPS data quality could be predicted by the mtDNA yield [30]. All tests were conducted using RStudio (2022.07.1+554, RStudio Team, 2022).

## 3. Results and Discussion

### 3.1. mtDNA Yield from Hair Shafts

The recovery of mtDNA from hair shafts (measured as copies of mtDNA/μL) was compared between two age categories [recent (R, <5 years) and old (O, 5–27 years)], both within and across the two hair cutting lengths (1 and 5 mm). Overall, mtDNA yield decreased significantly with age and length (Figure 2). When considering the 5 mm hairs, the R hairs had significantly more mtDNA recovered than the O hairs; *p*-value of 0.026 (Appendix A provides a list of *p*-values for mtDNA recovery comparisons). Comparing across the lengths, the R 5 mm hair extracts have significantly more mtDNA than all the 1 mm extracts; *p*-values of 4.67 × 10^−3^ for R hairs and 2.27 × 10^−8^ for O. The 5 mm O hair extracts also had a significantly higher mtDNA yield than the 1 mm O extracts; *p*-value of 4.82 × 10^−4^. This is consistent with the expectation that as more hair sample is extracted, more mtDNA is recovered and as samples age, less DNA is available due to factors such as degradation and damage [1].

Various types of DNA damage can lead to degradation and the inability for targets of interest to be amplified [31] impacting the analysis of mtDNA from hair shafts. The degradation index (DI) for hair extracts was determined by taking the ratio of the small to large target quantities of each quantification replicate and calculating the mean value for each individual hair sample, with statistical comparisons performed on the mean values. For samples where the large target value was undetermined, the quantity value was artificially set to the lowest datapoint in the standard curve of 10 copies/μL to allow a DI to be calculated. As expected, DIs increased significantly with age (Figure 3). Consistent with expectations, the O hairs had a significantly higher DI than the R of the same length; *p*-value of 0.034 for R 5 mm hairs (Appendix A provides a list of *p*-values for DI comparisons). No other comparisons regarding age or length were significant. In addition, of particular note, the O hairs have relatively low average DIs, suggesting that hairs up to 27 years of age do not generally experience significant DNA degradation. Contrary to expectations, when comparing different cutting lengths, the O 5 mm hairs have a higher DI than all 1 mm hairs, noting that the difference is not significant. With more than double the average DI when comparing the O 1 mm and 5 mm, this suggests that the range of template lengths may exhibit greater levels of degradation as mtDNA yields increase, while lower yields reflect templates of generally the same length. Overall, these findings are consistent with the expectation that older samples will have greater levels of degradation than recent samples, but that degradation levels may increase with increasing yields of mtDNA and hairs up to 27 years of age will still yield fairly good quality DNA.

The quantity and quality of mtDNA is highly variable in shed hairs from different individuals and even from hairs of the same individual [10,11,32,33]. In addition, previous studies have found that mtDNA content decreases distally from the root end of a hair [10]. Most of the hair cuttings in the current study were taken approximately 1 cm from the root end and should, therefore, reflect the greatest possible mtDNA yield when compared to the remaining hair shaft. However, a number of the hairs (approximately 29% of the R and O hairs) did not have a root or were re-extracted with additional cuttings taken at greater lengths from the proximal end. While the distance from the root may impact yield and DI, hair extracts in the current study yielded more copies of mtDNA than previous studies using similar and different extraction methods regardless of where the cutting was taken. For example, Brandhagen et al. [10] recovered 43,992 copies of the mitogenome/cm from a recent hair segment taken approximately 1 cm from the root end, while the average yield for the current study was almost 405,360 copies/cm, including a R hair cutting taken at an unknown distance from the root which yielded 333,523 copies/cm, and with an overall range of yields of 36,187 to greater than two million copies/cm (Appendix A, R and O hairs). Gallimore et al. [11] utilized the same extraction method except for the modifications made (Materials and Methods Section 2.3) and generated on average between 200–300 copies/μL for 5 mm cuttings of recently collected head hairs. In comparison, the same size cutting yielded on average 4289 copies/μL in the current study; Appendix A provides the quantities for all hairs in both copies/μL and copies/cm. These results demonstrate that the modified extraction method is highly robust. While many factors, including natural variability, the environment, and chemical or physical changes to the hair can impact mtDNA recovery, these results support that the age of the hair and the length of the cutting have the greatest impact on mtDNA yield.

### 3.2. Assessment of MPS Sequence Data Generated from Hair Shafts

A gross assessment of MPS data quality was linked to the number and types of minor sequence variants reported by the analysis software (GM HTS), the generation of complete haplotypes, and the coverage of reads across the mitogenome. Minor variants consist of mixed sites due to heteroplasmy, damage sites, possible contamination, NUMTs (nuclear inserts of mitochondrial DNA), or random error [17]. An elevated number of minor variants may indicate the presence of DNA damage. The number of minor variants called by GM HTS increased significantly as the age of the hairs increased (Figure 4). The O hairs have significantly more minor variants when compared to the R hairs within the two lengths (1 mm: *p*-value of 0.049; 5 mm: *p*-value of 8.59 × 10^−3^) and when comparing across the two lengths (*p*-value of 3.53 × 10^−4^) (See Appendix A for all *p*-values). Given that the number of minor variants and the DI relate to the amount of DNA damage in a sample, Figure 5 provides a comparison of the number of minor variants reported to the DI. For the R hairs, the DI and the number of minors are both low and cluster near the origin as they have less apparent damage. As age increases, the number of minor variants and the DI increase. For the O hairs, the number of minor variants increases more than the DI, resulting in a cluster along the y-axis. The adjusted R^2^ values in Figure 5 along with the residual standard error and F-statistic values presented in Appendix A, indicate that the DI is a good predictor for the number of minor variants for the longer, younger (R) hairs, but not the older (O) hairs. It was expected that as the DI increases, the number of minor variants would also increase as low template and low-quality samples have higher error and base substitution rates [29,31,34]. Due to the O hairs not showing much of an increase in the DI in comparison to the increase in minor variants, the amount of template DNA added to the reaction was assessed.

During amplification, the maximum volume of template DNA, 15 μL, was often used to get as close to 40,000 copies of input mtDNA. Due to the lower yield from the O 1 mm hairs, fewer copies were added to the reaction. Figure 6 shows that for the 5 mm O hairs, a higher DNA input resulted in fewer minor variant calls. The 1 mm O hairs follow a similar trend but with more hairs clustered at lower input amounts and a wider spread in the number of minor variants. The 5 mm R hairs are clustered near 40,000 copies and close to zero minor variants while the 1 mm R hairs are at a relatively constant number of minor variants with a widespread template input amount. It has been previously shown that damage sites tend to appear more when lower template amounts are used because those minor variants have more opportunities to be amplified when poorer quality DNA is added to the reaction [31]. The data in Figure 6 and Appendix A, illustrates that template amount may impact the number of minor variants when working with older samples; however, it is not a strong predictor. In general, inputting more DNA template does not mean the quality of the sequence results will be of better quality. If the DNA is damaged or degraded, adding more to the amplification reaction will still result in noisy sequencing results. When analyzing mtDNA MPS data, it is important to assess and understand noise or error rates. This will help to distinguish heteroplasmy from noise, as heteroplasmic sites are useful for increasing the discrimination potential of mtDNA sequences and can be used to distinguish between maternal relatives [29,35,36]. With the high number of minor variants being observed in the O hairs, assessment of low-level heteroplasmic sites becomes more challenging.

As stated in Section 2.1, the hairs provided by Microtrace were stored in plastic bags at the time they were collected (seven to 46 years ago). The hairs were dry when they were collected, put in polyethylene bags, and were then kept in file cabinets at ambient temperature. When comparing the MPS data generated from the hairs provided by Mitotyping (stored frozen) and Microtrace, the 12 O hairs from Mitotyping had significantly fewer minor variants than the 12 O hairs from Microtrace (*p*-value of 2.68 × 10^−3^). On average, the O Mitotyping hairs had 33.125 minor variants, the O hairs from Microtrace had 91.833 minor variants, and the 23 VO hairs from Microtrace had 207 minor variants called in GM HTS. Based on this comparison, it appears that the long-term storage in plastic impacted the MPS data quality in combination with the increasing age of the hairs.

When assessing the types of base changes for possible DNA damage observed in the MPS data, apparent deamination damage was identified. Cytosine deamination has been observed in forensic samples in previous studies [31,34,37,38]. Deamination events typically appear as C to T transitions on one strand or G to A transitions on the other. These base changes accounted for 57.9% and 23.6% of the minor base changes, respectively. The majority of these sites occurred in the O hairs (Figure 7), which is consistent with previous studies showing that older samples exhibit more DNA damage [39]. Chemical treatments to hair can also contribute to DNA damage. Microtrace provided hair treatment information for the donated hairs. There was a significant difference between hairs that were treated and not treated with respect to the percent of the whole mitogenome reported (*p*-value of 8.58 × 10^−3^), but no other aspects of the sequence quality or mtDNA yield experienced a significant difference (Appendix A). We predict that treated hairs will yield more damaged DNA, possibly caused by oxidative damage from bleaching and coloring and, therefore, partial mitogenome sequences. However, due to the low sample size of treated hairs (n = 8) and the high number of hairs where treatment is unknown (n = 48), more research is needed on the effects of hair treatment on mtDNA MPS data. An error assessment was performed on all 120 hair samples in the current study and is discussed below in Section 3.3.

In general, correct haplotypes were generated for all hair samples. While the actual haplotypes were unknown, as the donors were anonymous, no haplotype was observed more than once and when comparing the 1 and 5 mm cuttings from the same individual, the haplotypes matched over the comparable regions of sequence. Any differences were due to low coverage causing SNPs to drop out by not meeting the 200-read filter, being called minor SNPs, or not meeting the allele balance ratio filter (see Materials and Methods). Each haplotype was also run through Haplogrep2 to determine the haplogroup as extra quality control. The haplogroups matched for the 1 and 5 mm cuttings from the same individual except when key SNPs dropped out due to low coverage. The haplotypes and haplogroups are listed in Appendix A.

The minimum number of copies of mtDNA recommended to generate full mitogenome sequences is 20,000 copies. Therefore, hair extracts with at least 1333 copies of mtDNA/μL are expected to produce complete MPS profiles when considering a maximum input volume of 15 μL. Many of the older and shorter hairs yielded fewer mtDNA copies (Figure 2). Furthermore, the O hairs were also slightly more degraded (Figure 3). This combined data suggests that those hairs would have a lower success rate for generating full mitogenome sequences. Figure 8 further supports this expectation, illustrating that as the age of hairs increases, the percentage of the mitogenome reported at a read depth of greater than 200 reads per nucleotide position decreases significantly. Nonetheless, 44 of the 48 O samples resulted in full or almost full mitogenome sequences (98–100%). In turn, when the required read depth was lowered to 50, all 48 O hairs reached the 98–100% threshold, resulting in no significant differences due to age or length. At a read depth of 200, for the 1 mm, a higher percentage of the mitogenome is reported for the R hairs compared to the O hairs (*p*-value of 0.028). Additionally, the 1 mm O hairs had a significantly lower percentage reported than the 5 mm R hairs (*p*-value of 0.024; see Appendix A for all *p*-values). Overall, hair length does not appear to significantly impact the percentage of the mitogenome reported as much as the age since hairs of the same age, but different lengths did not significantly differ in the percentage of the whole mitogenome reported. On average, >99.5% of the mitogenome was reported at a read depth of greater than 200 reads per nucleotide when working with as little as 1 mm of hair. That average increases to 99.9% for a read depth of greater than 50 reads; noting that the haplotype does not change, it simply becomes more complete due to drop-out occurring when setting the filter to 200 reads. Therefore, crime labs can use a less stringent filter when reporting a haplotype if the profile is not complete. While this data does follow the trend of older samples not having as high of a success rate, full whole mitogenome profiles were still produced from almost all of the O 1 mm hairs. This supports that the MPS method is robust and can generate mitogenome sequences on a routine basis from as little as 1 mm of hair shafts up to 27 years of age.

Since the commercially available PowerSeq^TM^ Whole Mito System kit remains a prototype, the coverage of individual amplicons was evaluated to determine if there was a pattern of drop-out. Fifteen R and O samples experienced coverage dropping below 200 reads. Figure 9a plots read depths across the mitogenome for these 15 samples, while Figure 9b plots data for the entire set of 74 R and O samples tested. The data follows a similar trend in coverage and is comparable to previous assessments when sequencing the mitogenome [16]. Low-coverage zones are observed from nucleotide position (np) 1250–4000, np 7500–8000, np 9000–11,000, and np 15,400–16,000. Furthermore, the amplicon that spans np 2914–3034 dropped below 200 reads in 40% of the 15 samples, and amplicons spanning np 15,163–15,239 and np 8976–9073 dropped out in 47% of the 15 samples. This data suggests that certain regions are experiencing drop-out due to amplification and/or sequencing challenges, and as a result, consistently lower coverage can be expected in those regions.

### 3.3. MPS Error Assessment

As stated in the Materials and Methods section, 23 hairs ranging from 41 to 46 years old were also subjected to mitogenome MPS analysis. While the coverage and read depth were sufficient to generate at least partial haplotypes for each hair, they resulted in noticeably noisier sequences. McElhoe and Holland [29] previously generated an R script to characterize the background noise in MiSeq MPS data. This method was used to calculate the total error rate per sample and the top 25 highest positions of error per nucleotide for all 120 hair samples tested. Figure 10a supports the expectation that higher error rates are seen in older samples. This increase in error rate is statistically significant among all three age categories (Figure 10b; Appendix A) and aligns with expectations given qualitative assessments of the sequencing data. While the average error rate for each age category is relatively low, falling well below the minor variant reporting threshold of 2%, a higher average may translate into a greater number of individual nps where the noise exceeds the threshold.

Figure 11 provides an assessment of the top 25 nps with the highest rate of noise or error, on a per nucleotide basis (i.e., A, C, G, T) (see Appendix A for the top 25 positions and their frequencies). Statistical significance was calculated within each age category and across the same nucleotide (significant *p*-values for within each age category only are presented in Figure 11). The R and O hairs had significant differences in the frequency of noise between C and any other nucleotide; the VO hairs had significantly different error frequencies between C and G calls. A, G, and T calls were significantly higher in VO hairs compared to R hairs (see Appendix A for all *p*-values). These trends align with the paradigm that older hairs contain more damaged DNA which in turn is associated with noisier MPS data. The rate of A, G, and T error generally fell below the 2% reporting threshold, with certain sites above the threshold in the older hairs that appear as reportable mixed sites. The rate of C substitutions is generally the highest in all age groups and was above 2% in all age categories. The T error was the next highest, followed by A and then G error. The elevated C error rate can be attributed to a combination of the filtering limitations of the MiSeq with respect to A/C detection and the alignment and sequencing challenges associated with homopolymeric C stretches in the control region [26,40]. The higher rate of T error compared to A and G can be explained by the observed deamination damage (C > T mutations), which has been previously noted as a predominate error type in hair samples [29].

When sites of error exceed the minor variant reporting threshold, they complicate the interpretation of heteroplasmy. Substitutions caused by contamination, random error, and damage are typically random and will not be duplicated across replicates of the same sample [11,34,38]. Any duplicated sites when running replicates are more likely to be true heteroplasmy, excluding damage hotspots. Of the 300 nps represented in Figure 11 (25 nps per boxplot), 121 were observed once (40%) with 179 positions observed multiple times (60%). Of the 121 positions observed once, 30 sites (24.8%) were located in the control region of the mitogenome which is non-coding and contains the hypervariable regions [16]. The control region is 1122 nucleotides long and only represents 6.7% of the entire mitogenome (16,569 base pairs). The overrepresentation of error in the control region is expected given that mutations are more likely to occur in that region. Of the 300 error sites, 105 (35%) reached a frequency above the set 2% reporting threshold and 61 of those 105 (58%) were reported in at least one sample. A little more than half of the sites that were reported were in the VO hairs (55 sites; 52%). The remaining sites were not reported due to failing the other filter parameters including the read balance filter. The read balance filtering parameter is important as most hotspot error sites will not be reported due to imbalance [29].

When comparing the top error sites in the current study to the sites in hair samples from a previous error assessment [29], some of the same C error sites exceeded 2% (np 499, np 538, and np 545), but the other error type sites were not in common. Some of the sites for each nucleotide were duplicated across the different age groups, either being repeated in all three age groups or only in R and O or O and VO. Most of the duplicated sites were not reported, with only 31 sites exceeding all the filtering parameters. Therefore, as with other studies [29,31,38], increasing age and DNA damage will elevate the noise in MPS data, making it more difficult to interpret true heteroplasmy. Fortunately, recent hairs, which should be the majority of hairs received in crime laboratories, have low noise levels allowing for heteroplasmy to be detected and reported.

### 3.4. Microscopic Correlations

Microscopic characteristics such as hair width, medulla, and pigment were noted during examination of the hairs. It is expected that larger-diameter hairs will generally yield more DNA and more of a complete MPS profile [1,11]. In this data set there is an increase in mtDNA quantity with an increase in width, but it is not strongly correlated (Figure 12a; Appendix A). The linear regression model shows that there is a slight correlation for the longer and younger hairs and a poor correlation for the older hairs. Since the medulla is generally air space, if a medulla was present, its width was also subtracted from the width to determine if this changed the correlation to mtDNA yield. Certain data points shifted to the left (decrease in width), resulting in an increase in the R^2^ values for the 1 and 5 mm R hairs. However, the other R^2^ values decrease and the ability of the width to predict the mtDNA quantity remains the same (Figure 12b; Appendix A). To further assess this characteristic, the medulla was classified into four categories during microscopy: continuous, discontinuous, trace, and absent. The type of medulla did not significantly impact mtDNA yield, the DI, the number of minor MPS variants called, or the percent of the whole mitogenome reported (Appendix A). Therefore, while mtDNA yield increases with the hair’s width slightly, the presence or absence of a medulla does not appear to impact the mtDNA recovery or quality of the MPS result. This is contrary to previous contentions that the presence of a medulla would reduce the DNA content in a hair shaft [41].

The pigment level observed in the tested hairs included no pigment (albino), blonde, light brown, medium brown, dark brown, red, and dark auburn (Appendix A). When considering only the hair’s pigment, no trend was observed for the quantity and quality of recovered mtDNA or the quality of the MPS sequence data once the Holm correction was applied (Appendix A). The age and length of the hair had a more significant impact on the sequence results (See Section 3.1 and Section 3.2). A previous study found that hairs with darker pigments are more likely to produce a full mtDNA profile [1], which are not consistent with the results from this study. One of the hair samples from this study had no pigment (albino) and produced a complete mitogenome. Overall, data from the current study do not suggest that microscopical characteristics significantly impact mtDNA MPS results. Nonetheless, microscopical examination of hairs is still a necessary step in the analysis process to document the physical characteristics of the hair, including whether the hair is of human origin, when applicable, and whether the hair has been damaged. These data can be used to exclude hairs for DNA analysis, provide information on the individual’s ancestry, note any physical damage to the hair that could impact sequencing or provide details to a forensic investigation, and determine chemical treatment information such as bleaching or coloring.

## 4. Conclusions

Based on the findings of the current study, mitogenome sequences can be generated on a routine basis from as little as 1 mm of human hair shaft up to 27 years of age since time of hair collection, illustrating that the applied MPS method is highly effective and robust. Furthermore, data from 1 and 5 mm replicates gave the same haplotype illustrating that the applied MPS method is reliable. As expected, as age increases, yield decreases, degradation increases, and sequencing quality declines. While the total error rate was low, older hairs with greater DNA damage or random error resulted in a higher rate of minor MPS variants, especially for the 1 mm hairs, jeopardizing the ability to assess heteroplasmy in those hairs.

The time is now for employing MPS-based methods for mtDNA analysis of DNA recovered from human hairs collected in forensic cases. If implemented, this approach will result in complete mitogenome profiles for samples historically deemed too challenging to analyze and will contribute to solving more forensic cases. In addition, generating mitogenome profiles from hair samples will assist the work of anthropologists and medical geneticists when studying ancestral origins, migration patterns, and health-related conditions.

## Figures and Tables

**Figure 1 genes-13-02144-f001:**
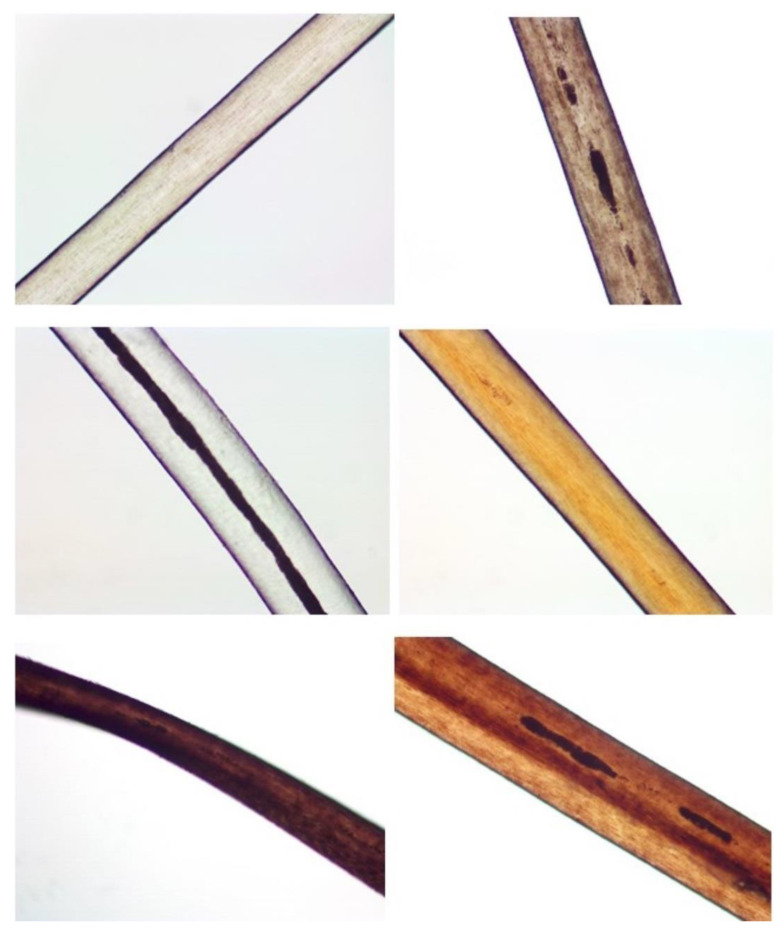
Pictures of representative hair samples. Moving down from left to right: R_03F_A, O_16_A, O_24_A, R_13_A, R_08F_A.3, and O_20_A. All pictures were taken at 200× magnification. R_08F_A.3 has an undulation and the donor of O_24_A has albinism.

**Figure 2 genes-13-02144-f002:**
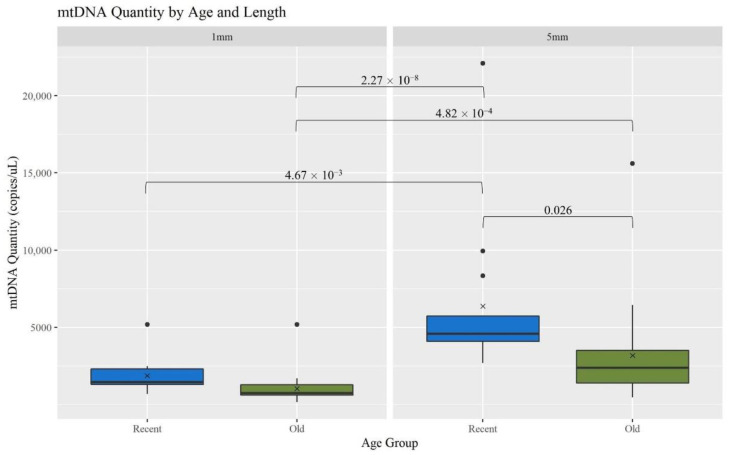
Boxplots for the comparison of mtDNA quantity for each age group and length (n = 74; n_R_ = 26, n_O_ = 48). The ‘x’ on the boxplots represents the mean and the line represents the median. Brackets show comparisons with significant *p*-values (*p* < 0.05).

**Figure 3 genes-13-02144-f003:**
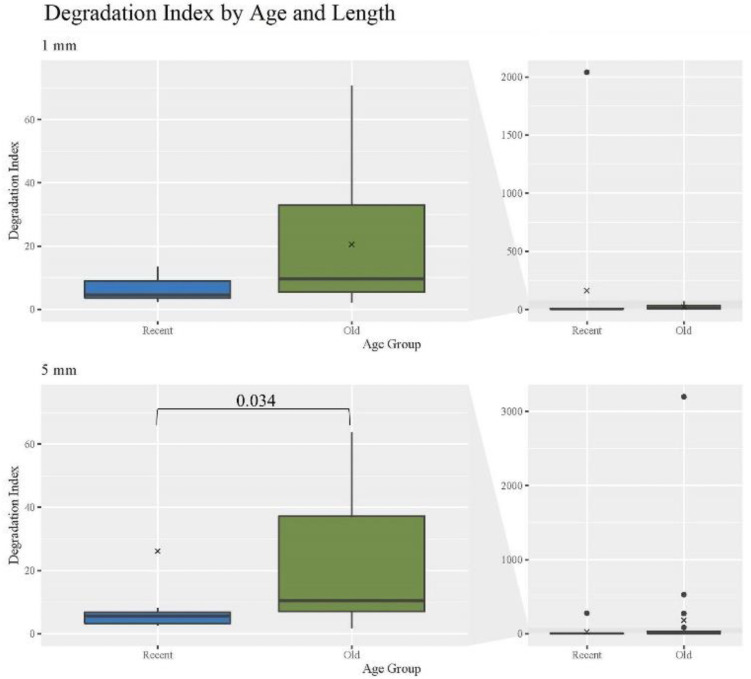
Boxplots for the comparison of degradation index for each age group and length (n = 74; n_R_ = 26, n_O_ = 48). The ‘x’ on the boxplots represents the mean and the line represents the median. Bracket shows comparison with significant *p*-value (*p* < 0.05).

**Figure 4 genes-13-02144-f004:**
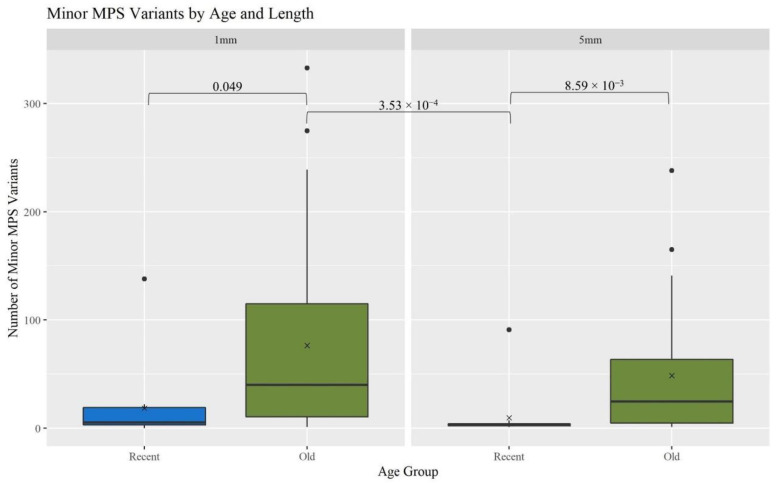
Boxplots for the comparison of the number of reported minor MPS variants for each age group and length (n = 74; n_R_ = 26, n_O_ = 48). The ‘x’ on the boxplots represents the mean and the line represents the median. Brackets show comparisons with significant *p*-values (*p* < 0.05).

**Figure 5 genes-13-02144-f005:**
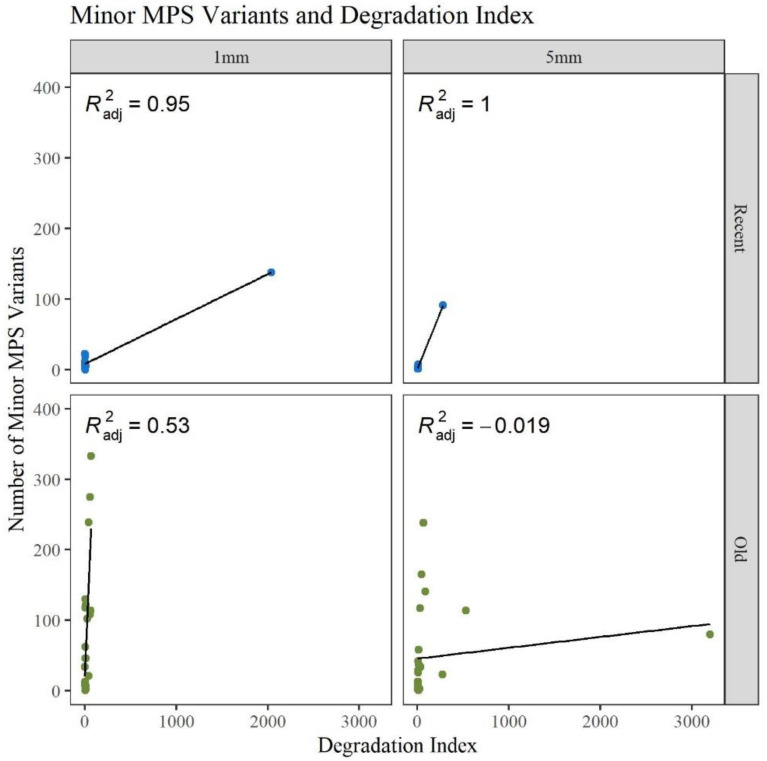
Linear regression of the number of reported minor MPS variants compared to the calculated degradation index for each age group and length (n = 74; n_R_ = 26, n_O_ = 48).

**Figure 6 genes-13-02144-f006:**
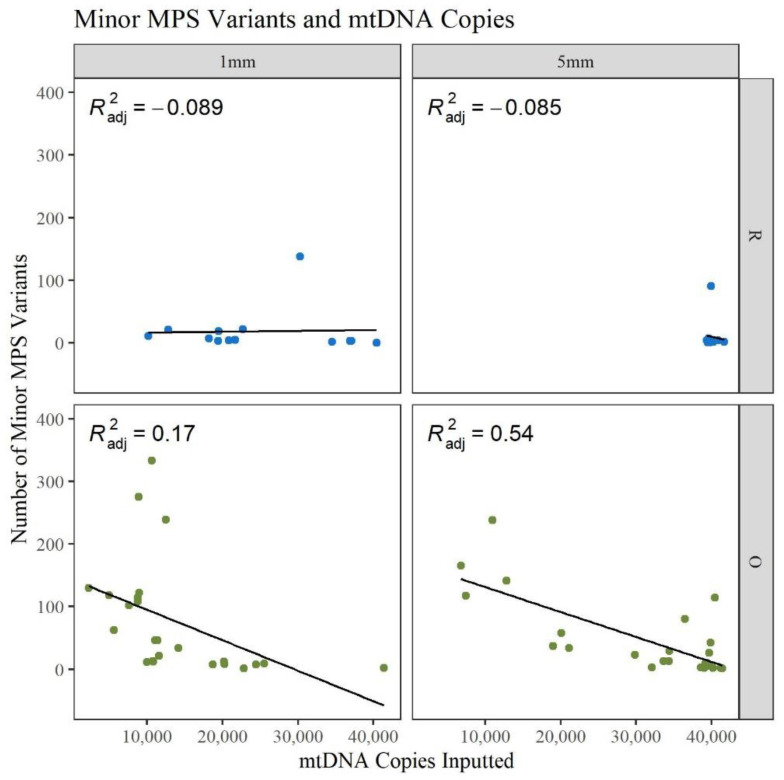
Linear regression of the number of reported minor MPS variants compared to the number of mtDNA copies inputted to the amplification reaction (n = 74; n_R_ = 26, n_O_ = 48).

**Figure 7 genes-13-02144-f007:**
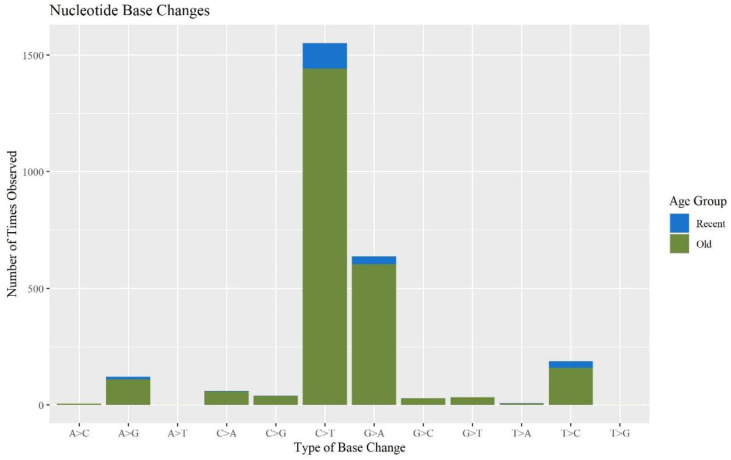
The number of times each base change is observed in the minor variants for each sample separated by age group (n = 74).

**Figure 8 genes-13-02144-f008:**
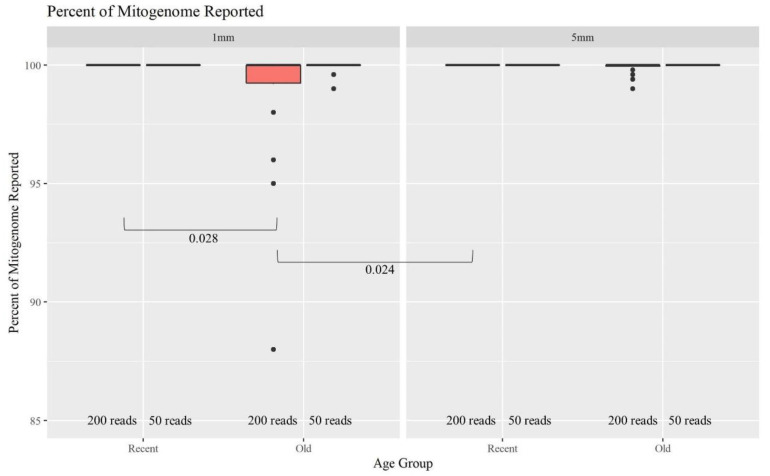
Boxplots for the comparison of the percent of the whole mitogenome reported for each age group and length at 200 reads and 50 reads (n = 74; n_R_ = 26, n_O_ = 48). The line on the boxplots represents the median. Brackets show comparisons with significant *p*-values (*p* < 0.05).

**Figure 9 genes-13-02144-f009:**
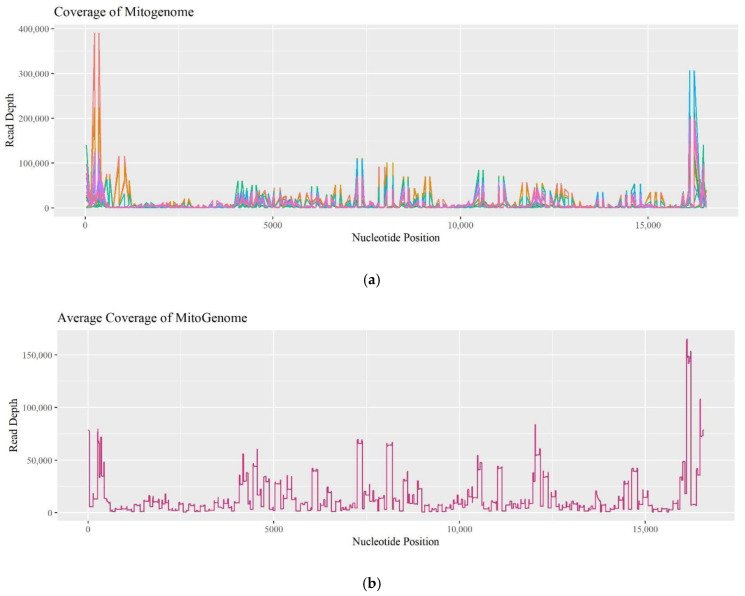
(**a**) A linear plot of the read depth across the mitogenome for the 15 samples that experienced amplicons dropping below 200 reads. (**b**) A linear plot of the average read depth for all 74 recent and old samples across the mitogenome.

**Figure 10 genes-13-02144-f010:**
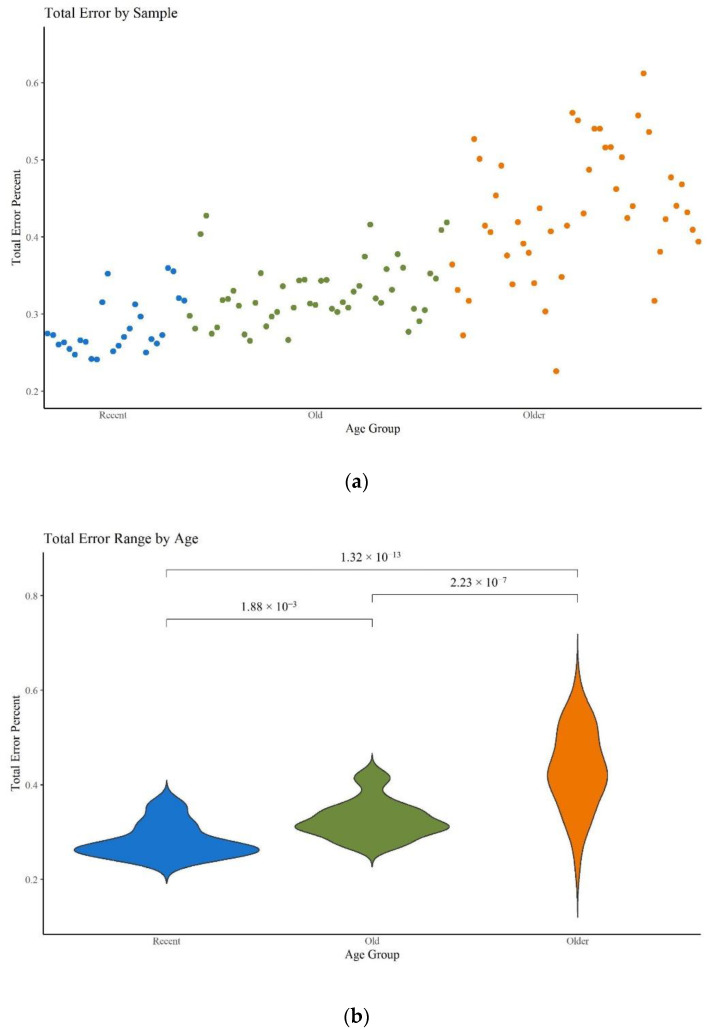
(**a**) Scatter plot of the total error for each sample. (**b**) Violin plot of the total error per age group. Brackets show comparisons with significant *p*-values (*p* < 0.05).

**Figure 11 genes-13-02144-f011:**
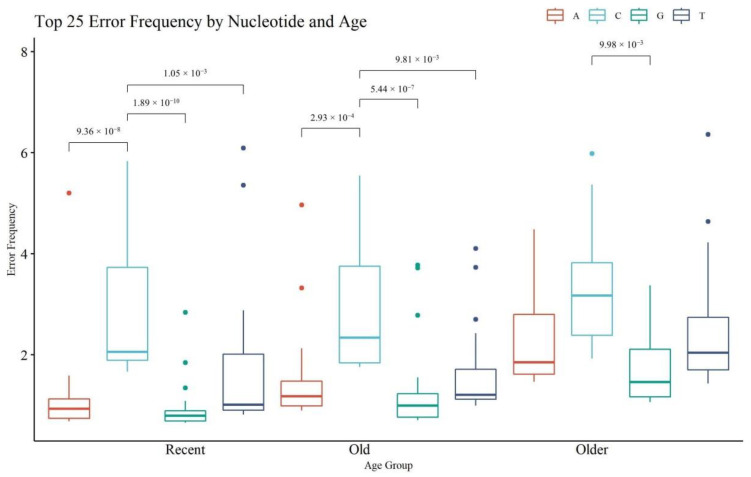
Boxplots of noise for the top 25 nucleotide positions, per nucleotide type (A, C, G, T), and per age. Brackets show comparisons with significant *p*-values (*p* < 0.05) within each age category.

**Figure 12 genes-13-02144-f012:**
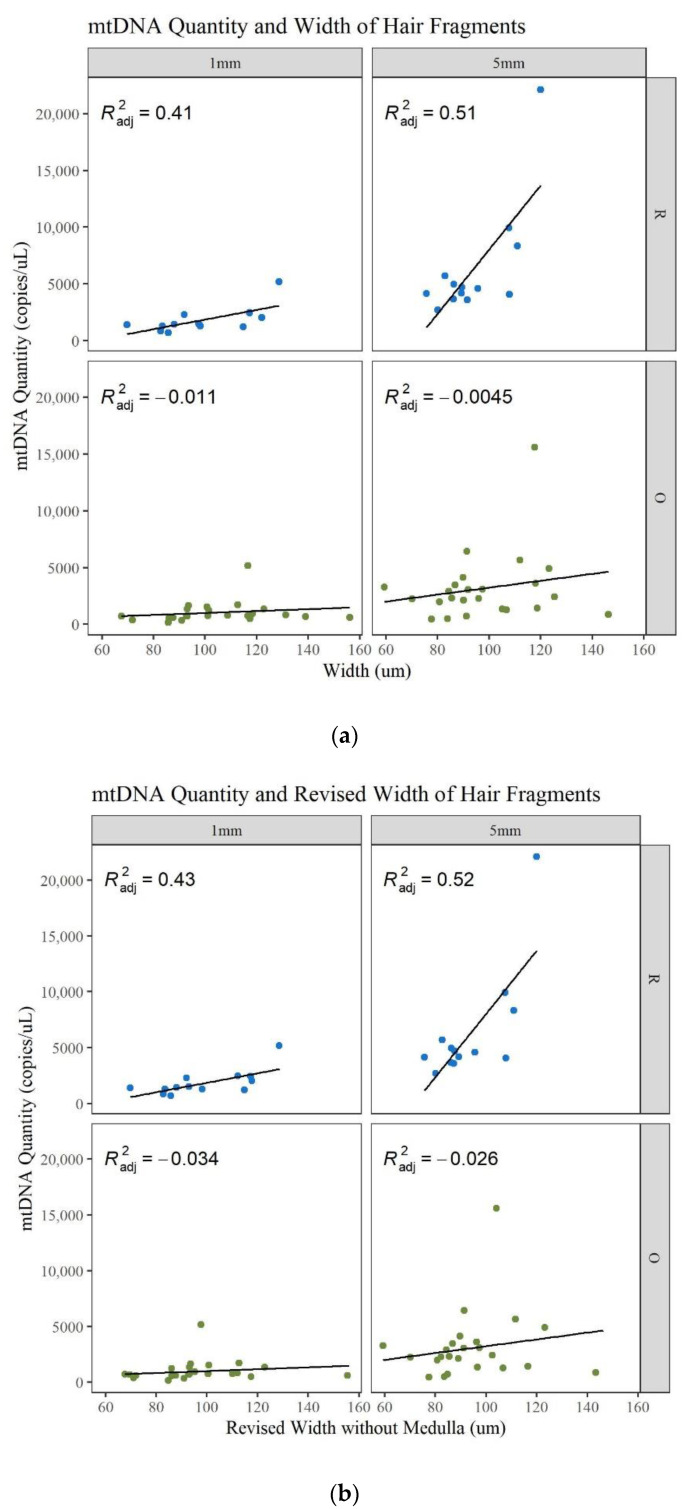
Linear regression of the mtDNA quantity compared to the width of the hair. (**a**) Width including the medulla. (**b**) Width excluding the medulla.

## Data Availability

The data presented in this study are available on request from the corresponding author. The data are not publicly available due to restrictions regarding consent.

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
