# Peer review of "Routine Mitogenome MPS Analysis from 1 and 5 mm of Rootless Human Hair"

_genes, 2022, doi:10.3390/genes13112144_

Round 1

Reviewer 1 Report

The authors present a study on hair shafts. For this purpose, they microscopically examined hair shaft sections of 1 and 5 mm in length from hairs less than five years to 46 years old from 60 donors and noted hair age (time since collection), hair diameter, medullary structure, and length of the tested hair. Then, then the DNA content was quantified and the entire mitochondrial genome of the hairs was sequenced using massively parallel sequencing (MPS).

The results show that reportable mitogenome sequences could be obtained up to 27 years of age. With increasing age of the hair, mtDNA degradation enhanced and mtDNA yield and mitogenome MPS data quality reduced. Hair diameter and mark structure had minimal effect on yield and data quality. Overall, this study demonstrates that MPS is a robust and reliable method for routine generation of mitogenome sequences from 1- and 5-mm hair shaft samples up to 27 years of age

The manuscript is written very well and comprehensibly. The literature is many and up-to-date. The introduction gives a very good overview of forensic molecular biology work with hair.

Very important and illustrative results are obtained. Especially the comparison between the microscopic characteristics of the different hairs and the results of the sequence analysis is very interesting. Furthermore, Figure 6 clearly shows that, especially for old samples, the number of templates can influence the number of minor variants. It is also shown that long-term storage in plastic compared to frozen hair affects the quality of the MPS data. The analysis of the error rates in the different hair categories in connection with the type and position of the bases is also very interesting.

The manuscript supports very well the thesis, also put forward by the authors, that it is time to use MPS-based methods for mtDNA analysis of DNA from human hair, as it is a robust and informative method. 

I regret that the analysis of very old hairs was removed from the manuscript. But I am very excited about the following paper.

In my opinion, this is an excellent manuscript. I strongly recommend its acceptance for publication. I have very few and small comments:

Minor point:

Table S1_ please explain the abbreviation "unk" in the legend

Figure 3: In my opinion, Figure 3A is redundant and has no added value compared to Figure 3b. So you could turn Figure 3b into Figure 3 and delete Figure 3A.

Reviewer 2 Report

the work is well designed and presented. none suggestions for the authors